# Position: Text Embeddings Should Capture Implicit Semantics, Not Just Surface Meaning

**Yiqun Sun** [1]   **Qiang Huang** [2]   **Anthony K. H. Tung** [1]   **Jun Yu** [2]

## Abstract

**This position paper argues that text embedding research should move beyond surface meaning and embrace implicit semantics as a central modeling objective.** Text embeddings are a foundational component of modern NLP, underpinning a wide range of applications and driving sustained research progress. Despite rapid progress, most embedding models remain narrowly focused on surface-level semantics, whereas linguistic theory emphasizes that much of human meaning is implicit, shaped by pragmatics, speaker intent, and sociocultural context. Current models are typically trained on datasets that lack such depth and evaluated using benchmarks that reward surface similarity. As a result, they struggle with tasks that require interpretive reasoning, stance recognition, or socially grounded understanding. Our pilot study makes this limitation explicit, showing that even state-of-the-art embeddings achieve only marginal improvements over simple lexical baselines on tasks probing implicit semantics. We therefore call for a paradigm shift: embedding research should prioritize linguistically grounded and diverse training data, develop benchmarks that probe deeper semantic understanding, and treat implicit meaning as a core modeling objective to better align embeddings with real-world language complexity. The code is available at `http://github.com/dukesun99/Implicit-Embeddings`.

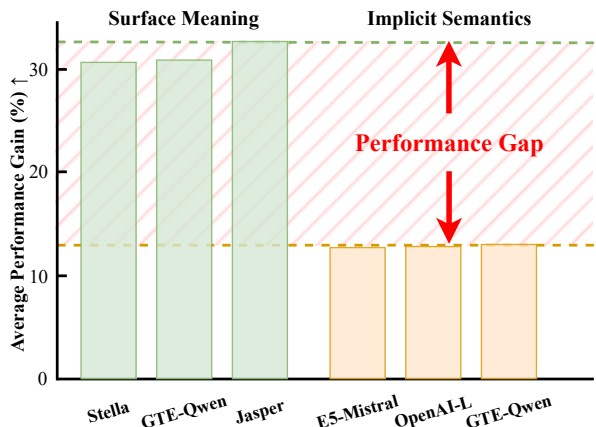

*Figure 1.* **Average gains over a lexical baseline (Bag-of-Tokens) for SOTA embedding models on Surface Meaning versus Implicit Semantics.** Surface performance is averaged over MTEB classification tasks (Muennighoff et al., 2023), while implicit-semantics performance is averaged over seven datasets probing pragmatic, attitudinal, and social meaning (Table 1). The comparison illustrates a performance divide: modern embeddings show large gains on surface-semantic benchmarks but much smaller gains on tasks requiring implicit interpretation.

## 1. Introduction

Text embedding models map sentences, passages, or documents into dense vectors whose proximity reflects semantic similarity (Reimers & Gurevych, 2019; Muennighoff et al., 2023; Sun et al., 2025d). They underpin much of modern NLP and Information Retrieval, serving as foundational components in Clustering (Grootendorst, 2022; Huang et al., 2023b; Angelov & Inkpen, 2024; Li et al., 2026), Classification (Muennighoff et al., 2023; Enevoldsen et al., 2025), Dense Retrieval (Thakur et al., 2021; Karpukhin et al., 2020; Sun et al., 2024; Huang et al., 2024; Tang et al., 2025), and Retrieval-Augmented Generation (RAG) (Lewis et al., 2020; You et al., 2026a;b; Dai et al., 2026; Sun et al., 2026). As a result, embedding models are widely deployed in a pre-trained, off-the-shelf manner and treated as general-purpose semantic interfaces for downstream decision-making.

This central role has driven rapid progress across model architectures (Reimers & Gurevych, 2019; Li & Zhou, 2024; BehnamGhader et al., 2024), training objectives (Thirukovalluru et al., 2024; Li & Li, 2024a; Xianming et al., 2025), and large-scale evaluation benchmarks (Muennighoff et al., 2023; Han et al., 2025; Enevoldsen et al., 2025). By stan-

---
[1]National University of Singapore [2]Harbin Institute of Technology (Shenzhen). Correspondence to: Qiang Huang <huangqiang@hit.edu.cn>.

*Proceedings of the 43rd International Conference on Machine Learning*, Seoul, South Korea. PMLR 306, 2026. Copyright 2026 by the author(s).

dard benchmarks, modern embedding models appear increasingly strong, robust, and general-purpose.

**The Overlooked Dimension: Implicit Semantics** Despite this progress, we argue that contemporary embedding research remains narrowly focused on surface meaning, such as semantic signals derived from lexical overlap, syntactic alternation, and topical similarity, while systematically underrepresenting the implicit semantics that are fundamental to human language understanding. Decades of linguistic research demonstrate that meaning is frequently conveyed indirectly, shaped by pragmatic inference, speaker intent, stance-taking, and sociocultural context rather than explicit propositional content (Huang, 2017; Ma et al., 2025; Kiesling, 2022; Silverstein, 2003; Bucholtz & Hall, 2005). Such implicit meanings determine not only what is said, but how and why it is understood in context.

Crucially, these dimensions of meaning are not peripheral edge cases. They govern everyday interpretation in domains such as persuasion, ideology, politeness, sarcasm, safety, and social signaling. Yet they remain largely invisible to embedding models optimized for surface-level similarity.

**Why Embedding Models Miss Implicit Meaning** This limitation is structural rather than incidental. First, dominant training corpora provide little supervision for implicit semantics. Most embedding models are trained on retrieval, entailment, or paraphrase datasets (Bajaj et al., 2016; Kwiatkowski et al., 2019), where success is defined by lexical relevance or literal semantic equivalence rather than contextual interpretation. Second, evaluation benchmarks overwhelmingly reward surface alignment. Widely used suites rarely test whether embeddings distinguish implied intent, speaker stance, or socially grounded meaning (Thakur et al., 2021; Muennighoff et al., 2023).

As a result, embedding models are optimized for what is easy to measure rather than what is linguistically consequential. Even advanced embeddings, despite high benchmark scores, are neither trained nor evaluated to capture the deeper layers of meaning that humans routinely infer.

**A Performance Divide** To examine this gap empirically, we conduct a pilot study spanning three tiers of implicit semantics: (1) utterance-level pragmatic inference, (2) speaker-level stance, and (3) society-level political and social meaning. Across a suite of linguistically motivated datasets, we find that leading embedding models perform only marginally better than sparse lexical baselines when implicit understanding is required.

Figure 1 summarizes this contrast. While modern embeddings achieve substantial gains over Bag-of-Tokens representations on surface-meaning benchmarks, their improvements nearly collapse on implicit semantics tasks. This performance divide highlights a fundamental mismatch between benchmark success and interpretive competence.

**Our Position** We argue that text embedding research must move beyond surface-level semantics and treat implicit meaning as a first-class modeling objective. This shift is essential for applications where semantic similarity alone is insufficient, such as stance-aware retrieval, ideological differentiation, or safety-critical filtering of content that appears benign on the surface but is implicitly harmful.

Our position is not to replace existing paradigms, but to broaden the community's understanding of what embeddings should represent. Aligning modeling objectives with linguistic theory and real-world interpretive demands can better capture the complexity of human communication.

## 2. Linguistic Foundations of Implicit Meaning

To clarify what we mean by *implicit meaning*, we draw on linguistic theory and organize it into a three-tier framework spanning the utterance, speaker, and society levels. We emphasize that this framework is intended as an analytical lens for organizing NLP-relevant phenomena, rather than a strict ontology or exhaustive linguistic definition of implicit meaning. This framework highlights how meaning systematically extends beyond literal content, emerging from pragmatic inference, speaker positioning, and sociocultural context.

**QUESTION** How do linguistic theories explain meaning that is conveyed but not explicitly stated?

**Utterance Level: Pragmatic Sources of Implicit Meaning** Pragmatics investigates how utterances derive meaning from context, bridging the gap between literal surface semantics and a speaker's intended message (Grice, 1975; Huang, 2017; Ma et al., 2025). Rather than focusing solely on what is explicitly said, pragmatics emphasizes what is left unsaid yet reliably understood, revealing interpretive layers that surface-level semantic analysis cannot fully capture. This view has long influenced NLP research on tasks that require contextual reasoning and inference (Hovy & Yang, 2021; Cambria, 2024).

A central insight of pragmatics is that meaning emerges from shared background knowledge, social norms, and situational context, which jointly constrain interpretation (Huang, 2017; Ma et al., 2025). Within this framework, speakers frequently rely on *implicature*–meanings inferred indirectly rather than stated outright (Grice, 1975; Potts, 2015; Hoyle et al., 2023; Ma et al., 2025). For instance, the sentence "*Bart managed to pass the test*" implies that his success was unexpected, even though this is not logically entailed.

Another key mechanism is *presupposition*, where utterances embed background assumptions required for comprehension (Potts, 2015; Ma et al., 2025). A statement like "*Sam quit smoking*" presupposes that Sam previously smoked, an

assumption that persists even under negation or questioning. Together, these phenomena illustrate how meaning depends not only on explicit content but also on what listeners infer or take for granted, posing a fundamental challenge for text embeddings that aim to model such nuance.

**Speaker Level: Stance and Implicit Meaning** While pragmatics focuses on utterances in context, *stance* focuses on the speaker's internal positioning: attitudes, evaluations, and degrees of alignment or commitment (Kiesling, 2022). Stance-taking is central to implicit meaning because it conveys emotional and social orientation through subtle linguistic cues rather than explicit statements.

Linguistic research characterizes stance along three dimensions: evaluation (positive or negative appraisal), alignment (social positioning relative to others), and investment (the strength of speaker commitment) (Du Bois, 2008; Lempert, 2008). These dimensions are often expressed through sociolinguistic variation. For instance, forms such as "*-in*" versus "*-ing*" function not merely as dialectal variants, but as signals of informality, toughness, or solidarity (Kiesling, 2009; Trudgill, 1972). Over time, such forms may become detached from specific groups and reused more broadly to index stance. The evolution of the word "*dude,*" from a gendered term to a marker of casual camaraderie, exemplifies this process (Kiesling, 2004).

Importantly, stance is dynamic. Quantitative studies show that speakers shift stance across discourse as they adjust intent and alignment (Kiesling et al., 2018). This introduces a relational and affective layer of meaning that complements pragmatics but remains difficult for embedding models to capture, as it depends on speaker intent rather than propositional content alone.

**Society Level: Sociocultural Foundations of Implicit Meaning** Beyond individual utterances and speakers, sociolinguistics examines how meaning is shaped by identity, power, and culture. Variation in pronunciation, grammar, or vocabulary–such as dropping the "*g*" in "*workin'*," regional vowel shifts, or particles like "*lah*" in Singapore English–functions as a social index, signaling class, group membership, or regional identity (Silverstein, 2003). These signals are culturally contingent: the same linguistic form may index friendliness in one context and stigma in another.

Language ideologies further shape implicit meaning by privileging certain varieties while marginalizing others (Bourdieu, 1991). Because high-status registers dominate most pretraining corpora, embedding models risk encoding and amplifying existing social hierarchies–for example, by systematically disadvantaging African-American Vernacular English relative to Standard English. Speakers also engage in style-shifting, alternating registers, dialects, or slang to negotiate identity and social relationships (Bucholtz & Hall,

2005). These shifts carry implicit meaning, signaling authority, solidarity, or deference.

Static embeddings, which average across usage, struggle to capture such rapid shifts of meaning. Capturing the societal dimension of language, therefore, requires sensitivity to culturally embedded cues beyond surface form.

**TAKEAWAY** Implicit meaning unfolds across three interconnected layers: (1) **pragmatics** captures what is implied but unsaid at the utterance level; (2) **stance-taking** reveals the speaker's evaluative and relational positioning; and (3) **sociolinguistics** exposes how language encodes identity, culture, and power. Together, these layers underscore that meaning is deeply contextual, socially embedded, and dynamically constructed, posing a significant challenge for text embeddings still anchored in surface-level representations.

## 3. Text Embedding Models: Landscape and Limitations

Text embedding–the task of mapping text into dense vector representations–has long been central to NLP and now underpins many state-of-the-art applications. This section reviews the evolution of embedding models, summarizes active research directions, and critically examines the field's current limitations. Figure 2 provides a high-level overview of major model classes and emerging trends that shape today's embedding landscape.

**QUESTION** What is the current state of research on text embedding models?

**Early Models** Early embedding approaches relied on static word vectors, such as Word2Vec and GloVe (Mikolov et al., 2013; Pennington et al., 2014), pooled into sentence-level representations. Subsequent models, including Skip-Thought (Kiros et al., 2015), InferSent (Conneau et al., 2017), Sent2Vec (Pagliardini et al., 2018), and the Universal Sentence Encoder (Cer et al., 2018), explore recurrent, transformer, or bilinear architectures to better capture sentence-level semantics. ELMo (Peters et al., 2018) marks a shift toward contextualized embeddings, dynamically encoding word meaning based on surrounding context.

**Encoder-Only Models** Pretrained encoder-only Transformers, such as BERT (Devlin et al., 2019) and RoBERTa (Liu et al., 2019), further advanced sentence embedding by enabling context-aware representations via pooled token embeddings. These models are typically optimized using contrastive or denoising objectives (Zhang et al., 2025c). Building on this foundation, methods such as Sentence-BERT (Reimers & Gurevych, 2019), SimCSE (Gao et al., 2021), TSDAE (Wang et al., 2021), and E5 (Wang et al., 2022) substantially improved embedding quality through refined training strategies, better negative sampling, and

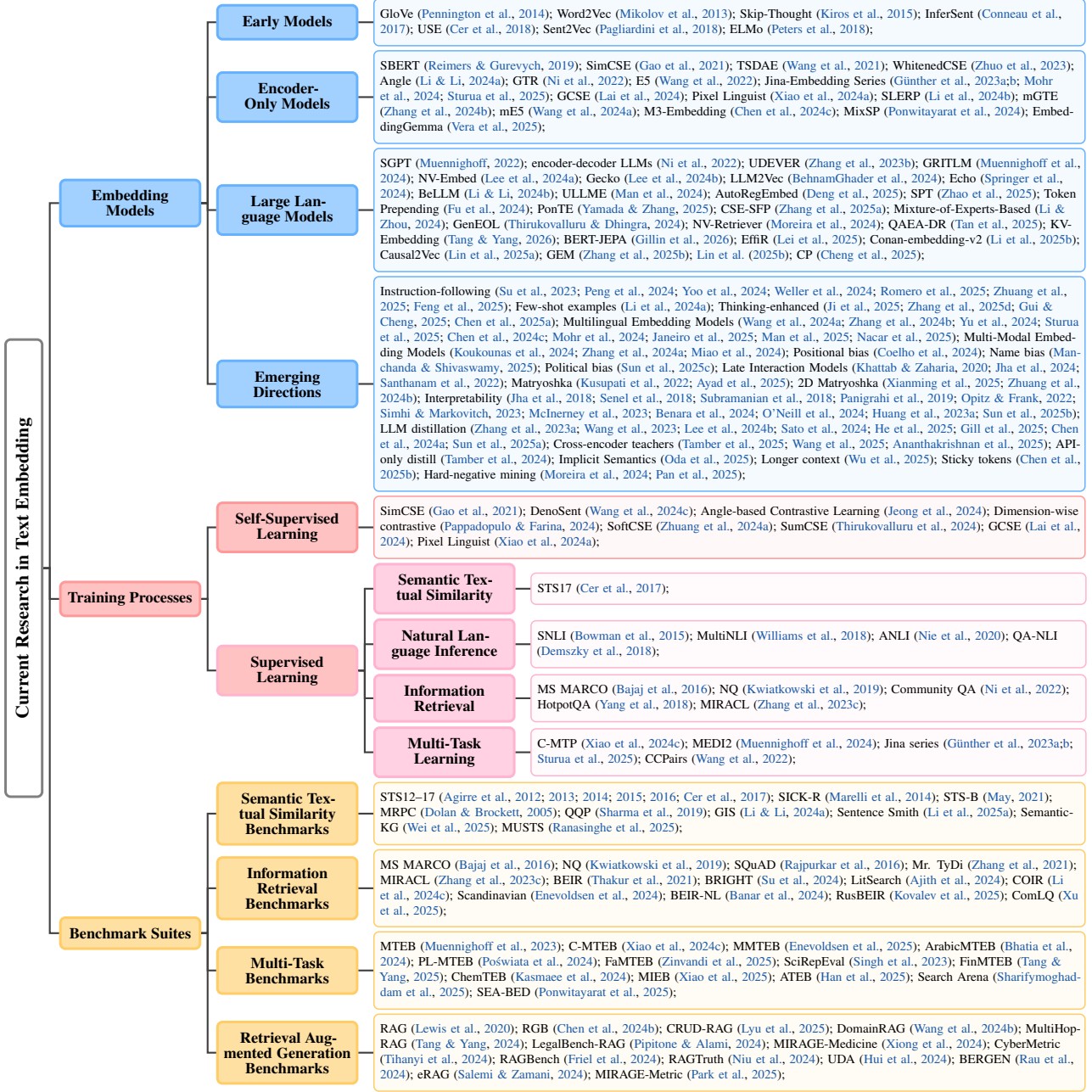

*Figure 2.* **A taxonomy of text embedding research:** tracing the evolution from early static models to recent trends in encoder-based architectures, LLM adaptations, multilingual embeddings, interpretability, and LLM distillation. While research has diversified, the modeling of implicit semantics remains underexplored.

architectural adjustments. As a result, encoder-only models remain the dominant choice in many retrieval and similarity-based applications due to their efficiency and strong benchmark performance (Zhuo et al., 2023; Koukounas et al., 2024; Lai et al., 2024; Xiao et al., 2024a; Li et al., 2024b; Ponwitayarat et al., 2024; Sturua et al., 2025; Chen et al., 2024c; Mohr et al., 2024; Günther et al., 2023a;b).

**LLMs as Embedders** More recently, LLMs have been adapted for embedding tasks using both decoder-only and encoder-decoder designs. Research has explored fine-tuning, prompting, and hybrid objectives to repurpose general-purpose LLMs for dense semantic representations. These efforts include adapting decoder-only models (Muennighoff, 2022; Cheng et al., 2025; Lin et al., 2025b), leveraging encoder-decoder architectures (Ni et al., 2022), or and building embedding-specific LLM variants (Muennighoff et al., 2024; Lee et al., 2024a;b; BehnamGhader et al., 2024;

Springer et al., 2024; Li & Li, 2024b; Man et al., 2024; Deng et al., 2025; Zhao et al., 2025; Fu et al., 2024; Yamada & Zhang, 2025; Zhang et al., 2025a; Li & Zhou, 2024; Zhang et al., 2025b). While such models often demonstrate strong semantic capabilities, converting generative LLMs into embedders typically requires adaptation with contrastive, retrieval, or similarity-based objectives, which may improve surface-semantic alignment without fully preserving the reasoning-relevant or implicit-semantic information available in the underlying model; their large size and high inference cost also sustain demand for lighter, encoder-based alternatives.

**Emerging Research Directions** Several trends characterize recent progress in text embeddings. Instruction-following (Su et al., 2023; Peng et al., 2024; Yoo et al., 2024; Weller et al., 2024; Feng et al., 2025; Zhuang et al., 2025; Romero et al., 2025), few-shot embedding (Li et al., 2024a), and "thinking-enhanced" representations (Ji et al., 2025) aim to improve adaptability across tasks. Multilingual and cross-lingual models (Wang et al., 2024a; Zhang et al., 2024b; Yu et al., 2024; Sturua et al., 2025; Chen et al., 2024c; Mohr et al., 2024) extend embedding utility beyond English, while new architectural designs address efficiency, political bias, and late interaction (Yoo et al., 2024; Coelho et al., 2024; Sun et al., 2025c; Khattab & Zaharia, 2020; Santhanam et al., 2022; Jha et al., 2024; Kusupati et al., 2022; Xianming et al., 2025; Zhuang et al., 2024b). Interpretability has also emerged as an important concern, with growing efforts to produce embeddings that align more closely with human-understandable concepts (Jha et al., 2018; Senel et al., 2018; Subramanian et al., 2018; Panigrahi et al., 2019; Opitz & Frank, 2022; Simhi & Markovitch, 2023; McInerney et al., 2023; Benara et al., 2024; O'Neill et al., 2024; Huang et al., 2023a; Sun et al., 2025b; Zhao et al., 2026).

Another prominent direction involves distilling knowledge from LLMs into lightweight embedding models. This includes mining hard negatives (Moreira et al., 2024; Pan et al., 2025), generating synthetic training data (Zhang et al., 2023a; Wang et al., 2023; Lee et al., 2024b; Sato et al., 2024; He et al., 2025; Gill et al., 2025), and transferring supervision from more accurate but slower teacher models (Tamber et al., 2025; Wang et al., 2025; Ananthakrishnan et al., 2025; Thirukovalluru et al., 2024; Tamber et al., 2024). While these techniques expand supervision and improve performance, they largely reinforce surface-level semantic alignment, with limited attention to implicit meaning.

**TAKEAWAY** Research on text embeddings spans architectures, training paradigms, multilinguality, interpretability, and efficiency. Yet, despite this rapid progress, the ability to capture implicit semantics–central to real-world language understanding–remains significantly underexplored. This gap motivates our subsequent analysis of training signals

and evaluation practices, our pilot empirical study, and the research agenda outlined in the following sections.

## 4. Training Processes Fail to Capture Implicit Semantics

Despite architectural advances, current training signals for text embeddings overwhelmingly supervise surface-level similarity rather than implicit meaning. This section examines the two dominant paradigms: self-supervised and supervised learning, and shows how both rely on datasets and objectives that privilege lexical overlap, paraphrastic equivalence, or relevance matching, leaving deeper pragmatic, attitudinal, and social meanings largely unmodeled.

**QUESTION** How are text embedding models trained, and why do these methods fail to capture implicit meaning?

### 4.1. Self-Supervised Learning

Self-supervised learning extracts training signals directly from unlabeled text using augmentation or structural cues, without requiring manual labels. Representative approaches include SimCSE (Gao et al., 2021), which creates positive pairs via dropout noise, and denoising-based objectives such as DenoSent (Wang et al., 2024c). More recent variants explore alternative formulations, including angle-based objectives (Jeong et al., 2024), dimension-wise contrastive loss (Pappadopulo & Farina, 2024), and similarity-weighted negative sampling (Zhuang et al., 2024a).

While self-supervised methods are attractive due to their scalability and low annotation cost, they consistently underperform supervised approaches in semantic benchmarks. As a result, most practical embedding pipelines adopt a two-stage strategy: self-supervised pretraining followed by supervised fine-tuning (Gao et al., 2021). Crucially, because the self-supervised signal is derived from surface perturbations of the same text, it primarily reinforces invariance to form rather than sensitivity to implicit intent or context.

### 4.2. Supervised Learning

Supervised embedding models use contrastive objectives such as triplet, SimCSE, or angle-based losses (Reimers & Gurevych, 2019; Gao et al., 2021; Li & Li, 2024a) on pretrained language models, but rely on task-specific datasets, primarily Semantic Textual Similarity (STS), Natural Language Inference (NLI), and Information Retrieval (IR), due to the scarcity of labeled pairs in general-purpose corpora (Raffel et al., 2020).

**Semantic Textual Similarity (STS)** STS datasets provide fine-grained similarity scores and have been widely used in early sentence embedding models (Reimers & Gurevych, 2019; Wang et al., 2021). However, their limited scale and

narrow domain coverage often lead to overfitting and weak generalization (Muennighoff et al., 2023). More importantly, STS annotations primarily reflect surface-level paraphrasing rather than deeper interpretive meaning.

**Natural Language Inference (NLI)** NLI datasets label sentence pairs as entailment, contradiction, or neutral and offer greater scale and diversity (Bowman et al., 2015; Williams et al., 2018; Nie et al., 2020; Demszky et al., 2018). They are widely used in modern embedding models (Reimers & Gurevych, 2019; Wang et al., 2021; 2022; Zhang et al., 2023b; Li & Li, 2024a; Zhang et al., 2023b). However, the semantic signal is often shallow: many entailment pairs differ only lexically or syntactically. For instance, pairs like "*A boy is jumping on a skateboard*" and "*The boy does a skateboarding trick*" test literal equivalence rather than pragmatic intent (Bowman et al., 2015).

**Information Retrieval (IR)** Retrieval datasets such as MS MARCO (Bajaj et al., 2016) and Natural Questions (NQ) (Kwiatkowski et al., 2019) dominate large-scale embedding training (Ni et al., 2022; Wang et al., 2022; Muennighoff et al., 2024; Moreira et al., 2024; Zhang et al., 2024b). These datasets are effective for learning lexical relevance and topic matching, but they reward literal overlap between queries and documents. As a result, embeddings trained on IR objectives struggle to encode implicit cues such as stance, ideology, sarcasm, or social framing.

**Multi-Task Learning** To improve robustness, recent models such as mGTE (Zhang et al., 2024b) and Jina (Günther et al., 2023a;b; Sturua et al., 2025) adopt multi-task training that combines STS, NLI, IR, QA, and related datasets (Xiao et al., 2024c; Muennighoff et al., 2024). While this broadens task and domain coverage, supervision remains largely surface-oriented, with few examples probing pragmatic inference, speaker stance, or sociocultural meaning.

**TAKEAWAY** Across both self-supervised and supervised paradigms, current embedding training pipelines are anchored in datasets that prioritize surface-level similarity. Although multi-task learning increases diversity, it does not fundamentally change what models are optimized to learn. As a result, core components of implicit meaning, i.e., pragmatics, stance, and social context, remain weakly represented or entirely missing from training objectives.

## 5. Benchmarks Do Not Evaluate Implicit Semantics

Despite the rapid expansion of large-scale benchmark suites–from semantic similarity and retrieval to multi-task generalization–most evaluations remain focused on surface-level semantics. This section surveys widely used STS datasets, retrieval, multi-task, and RAG benchmarks, highlighting a persistent gap: the limited evaluation of implicit, contextual, and socially situated meaning.

**QUESTION** How are text embedding models evaluated, and why do existing benchmarks fall short in capturing implicit meaning?

**STS Benchmarks** evaluate how well model-predicted similarities align with human judgments, typically using correlation-based metrics. Popular datasets include STS12–17 (Agirre et al., 2012; 2013; 2014; 2015; 2016; Cer et al., 2017), STS-B (May, 2021), and SICK-R (Marelli et al., 2014), along with related paraphrase classification tasks such as MRPC (Dolan & Brockett, 2005), QQP (Sharma et al., 2019), and GIS (Li & Li, 2024a). While STS tasks can probe deeper meaning, they are largely constrained to lexical and syntactic variation, rarely testing pragmatic, attitudinal, or cultural interpretation.

**IR Benchmarks** IR benchmarks evaluate how effectively embeddings retrieve relevant documents using ranking metrics such as MRR, nDCG, and recall@$k$ (Wang et al., 2013; Thakur et al., 2021). Datasets like MS MARCO (Bajaj et al., 2016), Natural Questions (Kwiatkowski et al., 2019), SQuAD (Rajpurkar et al., 2016), Mr. TyDi (Zhang et al., 2021), and MIRACL (Zhang et al., 2023c) are commonly used, with BEIR (Thakur et al., 2021) aggregating many such tasks across diverse retrieval settings. More recent benchmarks extend coverage to new domains and languages (Su et al., 2024; Ajith et al., 2024; Li et al., 2024c; Enevoldsen et al., 2024; Banar et al., 2024; Kovalev et al., 2025). Despite this breadth, IR benchmarks focus on surface relevance and seldom evaluate alignment with implicit criteria such as stance or ideology. Retrieval based on speaker stance or ideological framing remains largely unexplored.

**Multi-Task Benchmarks** MTEB (Muennighoff et al., 2023) seldom test whether retrieved documents align with implicit criteria such as stance, ideological framing, or communicative intent (Xiao et al., 2024c; Enevoldsen et al., 2025; Bhatia et al., 2024; Poświata et al., 2024; Zinvandi et al., 2025; Singh et al., 2023; Tang & Yang, 2025; Kasmaee et al., 2024; Xiao et al., 2025). More challenging variants introduce reasoning or instruction-following tasks (Han et al., 2025), and crowdsourced platforms like MTEB Arena[1] and Search Arena[2] provide user-driven comparisons (Sharifymoghaddam et al., 2025). Despite their scale and flexibility, these benchmarks rely largely on surface-level metrics and datasets. Only a small subset probes beyond lexical meaning, so strong MTEB performance often reflects surface alignment rather than sensitivity to implicit meaning.

**RAG Benchmarks** RAG benchmarks assess how well embeddings support retrieval for downstream generation.

---

[1] https://huggingface.co/spaces/mteb/arena
[2] https://blog.lmarena.ai/blog/2025/search-arena/

Existing benchmarks cover multilingual, domain-specific, and multi-hop scenarios (Chen et al., 2024b; Lyu et al., 2025; Wang et al., 2024b; Tang & Yang, 2024; Pipitone & Alami, 2024; Xiong et al., 2024; Tihanyi et al., 2024), and include tools for evaluating retrieval quality and hallucination (Friel et al., 2024; Niu et al., 2024; Hui et al., 2024; Rau et al., 2024; Salemi & Zamani, 2024; Park et al., 2025). While these setups introduce more complex pipelines, the underlying retrieval objectives remain largely factual. As a result, the underlying semantic evaluations resemble IR tasks, offering limited insight into how well embeddings reflect implicit intent, stance, or social meaning.

**TAKEAWAY**  Current benchmarks provide extensive task and domain coverage, but overwhelmingly emphasize surface-level similarity and relevance. They rarely assess a model's ability to capture implicit meaning, such as pragmatics, stance, or social context, leaving a critical gap in how we evaluate semantic understanding.

## 6. Empirical Evidences

To ground our argument empirically and motivate future research, we conduct a pilot study examining whether state-of-the-art embedding models capture implicit meaning across three linguistic tiers: utterance, speaker, and society.

**Experimental Setup**  We evaluate embeddings on seven datasets spanning three levels of implicit semantics: (1) Utterance level: Pragmatics Understanding Benchmark (PUB), including Implicature (**P-IMP**), Presupposition (**P-PRE**), and Reference & Deixis (**P-R&D**) (Sravanthi et al., 2024; Louis et al., 2020; Zheng et al., 2021; Liu et al., 2022; Chakrabarty et al., 2022; Jeretic et al., 2020; Parrish et al., 2021); (2) Speaker level: **P-Stance** dataset (Li et al., 2021) for stance detection; and (3) Society level: the datasets of Implicit Hate Speech (**IHS**) (ElSherief et al., 2021), Social Bias Inference Corpus (**SBIC**) (Sap et al., 2020), and Political Bias (**Pol. Bias**) (Baly et al., 2020). To provide a structured comparison, we report mean performance across subtasks within each semantic tier.

Since these datasets were not originally designed for embedding evaluation, we reformulate them into classification, pairwise classification (following the MTEB protocol (Muennighoff et al., 2023)), and zero-shot similarity settings, where models select labels based on embedding similarity. We evaluate four representative model families: (1) encoder-only models, (2) LLM-based embeddings, (3) multimodal encoders, and (4) proprietary embeddings (OpenAI). As baselines, we include a Bag-of-Tokens lexical model (Harris, 1954; Sun et al., 2025b) and a random predictor (Random) as a hardness baseline. Full implementation details are provided in Appendix A.

**Results and Analysis**  As depicted in Table 1, the results reveal a consistent and striking pattern. Encoder-only models typically perform only marginally better than the Bag-of-Tokens baseline and, in some cases, approach random performance. In contrast, LLM-based models and proprietary embeddings achieve stronger overall results. Notably, although OpenAI embeddings rank lower on standard benchmarks such as MTEB, they perform comparatively well on implicit semantics tasks, suggesting a disconnect between benchmark success and deeper semantic competence.

Performance also varies substantially across semantic tiers. Linq-Mistral excels at utterance-level pragmatic tasks, OpenAI-Large performs best on speaker- and society-level datasets, and E5-Mistral shows particular strength in political bias detection. These differences suggest that current models specialize unevenly across dimensions of implicit meaning, rather than learning a unified representation.

These positive results also clarify the nature of the limitation. Current embedding models are not uniformly poor on all implicit semantics tasks; rather, they perform better when the relevant implicit meaning is strongly associated with local lexical, syntactic, discourse, or label-specific cues. This is especially plausible for relatively conventionalized pragmatic phenomena, where large-scale pretraining and instruction tuning may help models recover recurring surface patterns correlated with the intended label. Thus, the issue is not complete failure, but uneven generalization: current embeddings appear to capture some shallow or partially lexicalized forms of implicit meaning, while remaining less robust on cases that require richer contextual grounding, speaker modeling, or socially situated interpretation.

These results support our central claim: **state-of-the-art embedding models remain limited in their ability to capture implicit semantics.** High MTEB scores do not translate into robustness on tasks involving pragmatic inference, stance recognition, or social meaning. The shrinking gains over Bag-of-Tokens, especially on the hardest pragmatic tasks, underscore a fundamental gap between how embeddings are evaluated and how meaning is actually constructed in language. Detailed results are provided in Appendix B.

## 7. Towards Embeddings that Capture Implicit Meaning

Building on the empirical findings in Section 6, we outline three directions for advancing text embeddings beyond surface-level semantics: (1) linguistically and culturally diverse training data, (2) benchmarks that directly evaluate pragmatic, attitudinal, and social understanding, and (3) treating implicit semantics as a core modeling objective. Together, these directions realign embedding research with the realities of human language interpretation.

*Table 1.* **Average accuracy (%) of representative embedding models** on seven datasets spanning three tiers of implicit semantics: utterance-level pragmatics, speaker-level stance, and societal-level social meaning. Results reveal substantial variation across semantic tiers and demonstrate that strong performance on standard benchmarks does not translate to robust modeling of implicit meaning.

| Model | Utterance Level | | | Speaker Level | Society Level | | | Average |
|---|---|---|---|---|---|---|---|---|
| | **P-IMP** | **P-PRE** | **P-R&D** | **P-Stance** | **IHS** | **SBIC** | **Pol. Bias** | **Accuracy ↑** |
| **Random** | 48.5 | 54.1 | 38.8 | 51.3 | 27.5 | 59.2 | 34.5 | 44.8 |
| **Bag-of-Tokens** (Sun et al., 2025b) | 56.5 | 75.3 | 48.2 | 73.4 | 59.6 | 80.7 | 41.6 | 62.2 |
| *Encoder-only Models* | | | | | | | | |
| **S-BERT** (Reimers & Gurevych, 2019) | 61.7 | 72.8 | 55.7 | 72.9 | 60.8 | 81.8 | 47.9 | 64.8 |
| **GIST-Small** (Solatorio, 2024) | 65.8 | 76.1 | 58.8 | 76.0 | 61.8 | 81.6 | 49.1 | 67.0 |
| **BGE-Base** (Xiao et al., 2024b) | 65.0 | 75.6 | 57.3 | 74.4 | 62.9 | 82.1 | 52.1 | 67.1 |
| **Angle** (Li & Li, 2024a) | 69.4 | 78.8 | 57.2 | 76.4 | 59.5 | 83.7 | 50.4 | 67.9 |
| **BGE-Large** (Xiao et al., 2024b) | 68.3 | 75.5 | 58.1 | 76.0 | 63.5 | 83.4 | 51.5 | 68.0 |
| **MXBAI-Large** (Lee et al., 2024c) | 69.8 | 78.2 | 59.4 | 75.6 | 60.1 | 83.6 | 50.5 | 68.2 |
| **GIST-Large** (Solatorio, 2024) | 68.8 | 76.6 | 62.9 | 76.7 | 64.2 | 83.4 | 52.5 | 69.3 |
| **Stella** (Zhang et al., 2024a) | 72.1 | 81.5 | 59.6 | 76.5 | 60.4 | 84.0 | 54.4 | 69.8 |
| *LLM-based Embeddings* | | | | | | | | |
| **Linq-Mistral** (Kim et al., 2024) | **80.3** | **87.7** | **70.4** | 75.8 | 61.4 | 82.0 | 56.8 | 73.5 |
| **E5-Mistral** (Wang et al., 2023; 2022) | 78.1 | 81.8 | 63.4 | 81.1 | 63.9 | 84.8 | **71.5** | 74.9 |
| **GTE-Qwen** (Li et al., 2023b) | 73.4 | 87.3 | 68.1 | 80.9 | 65.6 | 84.5 | 66.8 | **75.2** |
| *Multimodal Encoders* | | | | | | | | |
| **Jasper** (Zhang et al., 2024a) | 73.3 | 80.1 | 63.0 | 80.1 | 65.7 | 84.2 | 63.9 | 72.9 |
| *Proprietary Embeddings* | | | | | | | | |
| **OpenAI-Small** (Neelakantan et al., 2022) | 71.3 | 78.1 | 64.3 | 80.0 | 66.2 | 83.9 | 56.6 | 71.5 |
| **OpenAI-Large** (Neelakantan et al., 2022) | 76.0 | 80.2 | 66.4 | **83.7** | **67.1** | **85.4** | 66.3 | 75.0 |

## 7.1. Curating More Diverse Training Data

Training data ultimately determines what embedding models can learn. When supervision emphasizes surface-level signals, representations inevitably mirror surface meaning. Capturing implicit semantics, therefore, requires expanding beyond narrow, convenience-driven datasets toward linguistically richer and culturally diverse sources.

Importantly, by more diverse training data, we do not simply mean scaling raw web text. The key need is more diverse supervision for embedding learning. Unlike autoregressive LMs, embedding models are usually trained with pairwise, contrastive, retrieval-style, or classification-oriented signals. For implicit semantics, such supervision should include cases where surface meaning is similar but implied meaning differs, such as contrastive examples involving implicature, presupposition, stance, indirectness, sarcasm, social framing, dialectal variation, or ideology.

Recent progress in LLM-based data generation offers a practical path forward. Prior work shows that LLMs can synthesize effective supervision for embedding training, of-

ten by enforcing semantic invariance or contrast (Wang et al., 2023; Chen et al., 2024a; Lippmann & Yang, 2025; Truong et al., 2025). Future efforts should target implicit phenomena such as implicature, presupposition, stance, and indirect evaluation, rather than treating them as incidental variation. Such data can be curated from dialogue, indirect answers, stance-rich discourse, and socially situated language, or synthesized and relabeled using LLMs or stronger cross-encoder teachers guided by linguistic theory.

Linguistic theory provides a rich foundation for this shift. Decades of research have formalized typologies of implicit meaning across pragmatic and sociocultural dimensions. Aligning synthetic and curated data with these frameworks can help embedding models internalize forms of meaning that are largely absent from existing training corpora.

## 7.2. Designing Benchmarks for Implicit Meaning

Benchmarks shape research priorities by defining what success looks like. Existing suites, most notably MTEB, have played an important role in standardizing evaluation, but they overwhelmingly emphasize surface similarity. Their

open and leaderboard-driven nature has also led to data leakage and score inflation, weakening their ability to measure generalization (Chung et al., 2025; Sancheti et al., 2025).

Empirical studies show that strong MTEB results do not reliably translate to downstream robustness and may even correlate negatively with performance on certain tasks. This trend reflects a broader shift toward optimizing benchmark artifacts rather than transferable semantic understanding.

To address this gap, new benchmarks should explicitly target implicit meaning. This includes tasks that require inference from indirect cues, recognition of speaker stance, sensitivity to sociolinguistic variation, and interpretation of socially situated language. Without such benchmarks, progress on implicit semantics will remain invisible and undervalued.

### 7.3. Framing Implicit Semantics as a Modeling Goal

A deeper issue is that implicit meaning is rarely treated as a first-class objective in embedding research. While work on LLMs increasingly targets pragmatic reasoning, social understanding, and discourse-level interpretation (Li et al., 2020; 2023a; Kazemi et al., 2023; Sravanthi et al., 2024; Yue et al., 2024; Curry et al., 2024; Sun et al., 2024; 2025b; Ma et al., 2025), embedding models continue to be optimized for objectives that reward superficial alignment.

This misalignment pushes models to optimize what is easy to measure rather than what is meaningful to understand, reinforcing shallow representations in the absence of explicit objectives for implicit semantics. Reframing implicit semantics as a core modeling goal can better align embeddings with how humans interpret language in context.

Concretely, this goal can be operationalized through objectives that distinguish texts with similar explicit content but different implied meanings, multi-task supervision over pragmatics, stance, and social meaning, or distillation from models that explicitly reason over implicit interpretations. The goal is not to make embeddings preserve all information in the original text. Rather, it is to preserve task-relevant implicit signals that matter for downstream applications such as stance-aware retrieval, socially grounded classification, recommendation, clustering, filtering, and reranking.

Instruction-following retrieval is a particularly relevant step in this direction because it conditions embeddings on user intent and task framing rather than only surface similarity. Recent work on instruction-following embeddings and retrieval models (Su et al., 2023; Peng et al., 2024; Weller et al., 2024; Feng et al., 2025; Zhuang et al., 2025) shows that embedding research is already moving toward richer task-conditioned representations.

We view this line of work as a promising bridge between standard semantic embeddings and implicit-semantics-aware representations. Nevertheless, following retrieval instructions does not by itself guarantee sensitivity to broader implicit phenomena such as stance, presupposition, sociocultural meaning, or social indexicality.

## 8. Alternative Views

It is reasonable to argue that surface-level semantics suffice for many practical applications, such as search, recommendation, or clustering. In these settings, modeling deeper meaning may introduce unnecessary complexity and additional training or evaluation costs without clear returns. Another perspective holds that implicit, pragmatic, and socially grounded meaning is better handled by LLMs designed for contextual reasoning, while embeddings should prioritize efficiency and general-purpose utility. From this perspective, expanding the embedding scope risks blurring their role.

We acknowledge these positions. Our argument is not that embeddings should replace decoder-only LMs for complex reasoning, nor that all implicit understanding must be solved purely in vector space. Rather, embeddings often serve as first-stage representations in practical systems, where they determine which examples are retrieved, clustered, recommended, filtered, or passed to downstream models. In such settings, if important signals such as stance, intent, or social framing are not preserved in the embedding space, later components may never receive the relevant evidence.

Thus, as embeddings increasingly serve as semantic interfaces for downstream decision-making, their limitations in capturing implicit meaning become harder to ignore, especially in applications where stance, intent, or social interpretation are central.

## 9. Conclusions

Despite rapid progress, contemporary text embedding models are predominantly optimized for surface-level semantics, exhibiting a fundamental limitation in capturing the implicit meanings that are essential to nuanced human communication. Grounded in a three-tier linguistic framework: spanning utterance-level pragmatics, speaker-level stance, and society-level sociocultural context, our empirical analysis reveals that even state-of-the-art models show only marginal gains over simple lexical baselines on tasks probing these deeper layers of interpretation. These findings expose a structural mismatch between training, evaluation, and real-world language use. To bridge this gap, we advocate for semantically richer data, benchmarks that explicitly test implicit understanding, and modeling objectives that treat implicit semantics as a first-class goal. Aligning embedding research with these dimensions of real-world language use is crucial for developing robust, context-aware models that power the next generation of language-aware applications.

## Acknowledgements

Qiang Huang was supported by the New Generation Artificial Intelligence-National Science and Technology Major Project (2025ZD0123302) and the National Natural Science Foundation of China (NSFC) under Grant No. U25B6003. Jun Yu was supported by the NSFC under Grant Nos. 62125201 and U24B20174. Yiqun Sun and Anthony T. H. Tung were supported by the Ministry of Education, Singapore, under its MOE AcRF TIER 1 Grant (T1 251RES2517) and the National Research Foundation, Singapore under its AI Singapore Programme (AISG Award No: AISG3-RP-2022-029). Any opinions, findings, and conclusions or recommendations expressed in this material are those of the author(s) and do not reflect the views of the Ministry of Education, Singapore, or the National Research Foundation, Singapore.

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

# A. Implementation Details

## A.1. Model Checkpoints

We use the model checkpoints listed in Table 2 for all experiments reported in Section 6. To ensure a fair and reproducible comparison, we adopt the **official checkpoints** used by the MTEB benchmark and evaluate every open-source model under the **default configurations** provided by the Sentence Transformers library (Reimers & Gurevych, 2019). Importantly, we do not apply any additional parameter tuning, task-specific calibration, or prompt engineering; this design choice isolates the intrinsic capability of each embedding model under standard deployment settings.

**Proprietary Embeddings** For OpenAI's proprietary embedding models, we obtain representations using OpenAI's official client library[3] and follow the recommended embedding API usage. This ensures that the results reflect the intended inference behavior of these models rather than custom wrappers or ad hoc post-processing.

**Complementary Baselines** To contextualize performance, we include two complementary baselines:

- **Random:** For each dataset, we implement a random predictor by sampling labels according to the empirical label distribution. This provides a hardness reference that accounts for class imbalance and prevents overstating performance on skewed datasets.
- **Bag-of-Tokens** (Harris, 1954; Sun et al., 2025b)**:** As a lexical-feature baseline, we include a Bag-of-Tokens representation using the `google-bert/bert-base-uncased` tokenizer. This baseline serves as a strong surface-matching reference point, helping quantify how much modern embeddings improve beyond lexical overlap.

Together, these implementation choices are designed to make comparisons consistent across model families, while keeping the evaluation protocol aligned with widely used community standards.

*Table 2.* **List of evaluated embedding models and the corresponding checkpoints used in our experiments.**

| Model | Model Size | Checkpoint |
|---|---|---|
| **S-BERT** (Reimers & Gurevych, 2019) | 22.7M | `sentence-transformers\all-MiniLM-L6-v2` |
| **GIST-Small** (Solatorio, 2024) | 33.4M | `avsolatorio\GIST-small-Embedding-v0` |
| **BGE-Base** (Xiao et al., 2024b) | 109M | `BAAI\bge-base-en-v1.5` |
| **Angle** (Li & Li, 2024a) | 335M | `WhereIsAI\UAE-Large-V1` |
| **BGE-Large** (Xiao et al., 2024b) | 335M | `BAAI\bge-large-en-v1.5` |
| **MXBAI-Large** (Lee et al., 2024c) | 335M | `mixedbread-ai\mxbai-embed-large-v1` |
| **GIST-Large** (Solatorio, 2024) | 335M | `avsolatorio\GIST-large-Embedding-v0` |
| **Stella** (Zhang et al., 2024a) | 435M | `NovaSearch\stella_en_400M_v5` |
| **Linq-Mistral** (Kim et al., 2024) | 7.11B | `Linq-AI-Research\Linq-Embed-Mistral` |
| **E5-Mistral** (Wang et al., 2023; 2022) | 7.11B | `intfloat\e5-mistral-7b-instruct` |
| **GTE-Qwen** (Li et al., 2023b) | 7.61B | `Alibaba-NLP\gte-Qwen2-7B-instruct` |
| **Jasper** (Zhang et al., 2024a) | 1.99B | `NovaSearch\jasper_en_vision_language_v1` |
| **OpenAI-Small** | N.A. | `text-embedding-3-small` |
| **OpenAI-Large** | N.A. | `text-embedding-3-large` |

## A.2. Implicit Semantics Tasks

Our evaluation targets **implicit semantics** across diverse task formulations and data sources. Because the underlying datasets differ in structure and annotation format, we organize the evaluation into three settings: **classification**, **pair classification**, and **zero-shot classification**. Each setting includes the following datasets:

- **Classification:** From the **Pragmatics Understanding Benchmark (PUB)**, we include *Task 1 (Direct/Indirect Classification)*, *Task 2 (Response Classification without Implied Meaning)*, *Task 3 (with Implied Meaning)*, *Task 6 (Understanding Sarcasm)*, *Task 10 (Implicature NLI)*, *Task 11 (Presupposition NLI)*, *Task 12 (Presupposition over QA)*, and *Task 13 (Deictic QA)*. We also evaluate all three subsets of the **P-Stance** dataset–*Trump*, *Biden*, and *Bernie*–for stance classification. For the **Implicit Hate Speech (IHS)** dataset, we include *detection*, *categorization*, and *target identification* tasks.

---

[3] `https://platform.openai.com/docs/api-reference/embeddings`

*Table 3.* **Accuracy (%) of embedding models on the Pragmatics Understanding Benchmark (PUB) tasks.** Each column corresponds to one of the 14 PUB tasks (T1–T14), covering diverse pragmatic phenomena.

| Model | Pragmatics Understanding Benchmark (PUB) | | | | | | | | | | | | | |
| --- | --- | --- | --- | --- | --- | --- | --- | --- | --- | --- | --- | --- | --- | --- |
| | Implicature | | | | | | | | | | Presupposition | | Ref. & Deixis | |
| | T1 | T2 | T3 | T4 | T5 | T6 | T7 | T8 | T9 | T10 | T11 | T12 | T13 | T14 |
| **Random** | 48.0 | 49.2 | 52.9 | 24.9 | 50.0 | 51.0 | 49.2 | 47.1 | 49.7 | 62.9 | 36.4 | 71.7 | 56.0 | 21.7 |
| **Bag-of-Tokens** (Sun et al., 2025b) | 81.2 | 69.4 | 78.2 | 29.3 | 51.0 | 12.5 | 50.4 | 74.7 | 32.7 | 86.0 | 66.9 | 83.6 | 64.5 | 31.9 |
| *Encoder-only Models* | | | | | | | | | | | | | | |
| **S-BERT** (Reimers & Gurevych, 2019) | 82.0 | 72.4 | 83.1 | 35.4 | 53.5 | 34.0 | 59.9 | 78.6 | 39.0 | 79.0 | 60.0 | **85.6** | 68.5 | 42.9 |
| **GIST-Small** (Solatorio, 2024) | 74.8 | 74.2 | 83.3 | 44.5 | 56.1 | 40.2 | 67.4 | 86.8 | 52.4 | 77.9 | 66.9 | 85.2 | 68.5 | 49.0 |
| **BGE-Base** (Xiao et al., 2024b) | 83.0 | 71.5 | 79.6 | 33.2 | 56.0 | 42.8 | 69.8 | 80.9 | 55.6 | 77.6 | 66.1 | 85.2 | 68.0 | 46.6 |
| **Angle** (Li & Li, 2024a) | 97.2 | 77.3 | 81.2 | 41.5 | 58.1 | 32.8 | 75.1 | 82.8 | 60.7 | 87.6 | 74.2 | 83.4 | 66.0 | 48.4 |
| **BGE-Large** (Xiao et al., 2024b) | 87.0 | 76.8 | 79.8 | 43.0 | 57.7 | 41.8 | 75.3 | 84.0 | 59.6 | 77.9 | 65.8 | 85.2 | 68.0 | 48.1 |
| **MXBAI-Large** (Lee et al., 2024c) | 97.2 | 78.4 | 81.0 | 41.2 | 58.5 | 34.0 | 75.5 | 83.3 | 61.2 | 87.4 | 73.6 | 82.8 | 67.5 | 51.2 |
| **GIST-Large** (Solatorio, 2024) | 79.6 | 76.8 | 83.1 | 46.9 | 57.9 | 37.8 | 78.0 | 88.5 | 61.5 | 78.3 | 68.1 | 85.2 | 71.5 | 54.3 |
| **Stella** (Zhang et al., 2024a) | 95.8 | 79.8 | 81.9 | 51.1 | 60.0 | 35.5 | 76.2 | 87.7 | 60.8 | 91.7 | 83.6 | 79.3 | 66.0 | 53.2 |
| *LLM-based Embeddings* | | | | | | | | | | | | | | |
| **Linq-Mistral** (Kim et al., 2024) | 99.6 | **89.3** | **88.6** | 47.5 | **70.0** | 67.0 | **88.6** | **94.5** | 61.4 | **96.2** | **91.4** | 84.0 | 74.0 | **66.7** |
| **E5-Mistral** (Wang et al., 2023; 2022) | 97.2 | 87.2 | 85.8 | 43.8 | 69.5 | **69.0** | 87.7 | 92.7 | **62.9** | 85.0 | 78.3 | 85.2 | 68.0 | 58.9 |
| **GTE-Qwen** (Li et al., 2023b) | **100.0** | 87.7 | 87.5 | 43.6 | 61.8 | 63.0 | 69.0 | 82.7 | 48.5 | 89.8 | 89.4 | 85.2 | **78.0** | 58.2 |
| *Multimodal Encoders* | | | | | | | | | | | | | | |
| **Jasper** (Zhang et al., 2024a) | 97.8 | 84.2 | 85.4 | 50.6 | 61.6 | 49.8 | 78.0 | 88.4 | 55.4 | 81.9 | 75.0 | 85.2 | 70.5 | 55.6 |
| *Proprietary Embeddings* | | | | | | | | | | | | | | |
| **OpenAI-Small** | 98.8 | 79.4 | 84.9 | **56.0** | 56.7 | 35.5 | 79.3 | 89.9 | 55.0 | 78.1 | 71.1 | 85.2 | 73.0 | 55.6 |
| **OpenAI-Large** | 99.6 | 87.7 | 87.7 | 50.8 | 61.9 | 55.5 | 83.9 | 91.8 | 58.4 | 83.1 | 75.3 | 85.2 | 73.5 | 59.3 |

For the **Social Bias Inference Corpus (SBIC)**, we evaluate five binary classification tasks: *whoTarget* (whether the target is a group), *intentYN* (intent to offend), *sexYN* (presence of sexual content), *offensiveYN* (offensiveness), and *hasBiasedImplication* (biased implications). Lastly, we include the **Political Bias (Pol. Bias)** classification dataset.

- **Pair Classification:** We adapt *Task 5 (Agreement Detection)* from **PUB**.
- **Zero-shot Classification:** We include *Task 4 (Implicature Recovery)*, *Task 7 (Figurative Language Understanding–No Hint)*, *Task 8 (with Positive Hint)*, *Task 9 (with Contrastive Hint)*, and *Task 14 (Reference via Metonymy)* from **PUB**.

### A.3. Evaluation Protocols

For **classification** and **pair classification** tasks, we follow the standard evaluation protocol used in MTEB (Muennighoff et al., 2023). In this setting, embeddings are computed for inputs (and, when applicable, for candidate labels or verbalized label descriptions), and predictions are made using the corresponding embedding-based classification procedure consistent with MTEB's implementation.

For **zero-shot classification**, we follow the embedding-based approach[4] described in OpenAI's documented use case: the input query (or question) and its associated text are embedded jointly, while each candidate answer option is embedded separately. The model selects the option with the highest similarity to the input embedding. This protocol provides a standardized way to test "label selection by semantic similarity," allowing comparison across both open-source and proprietary embeddings.

## B. Additional Results

The complete results, including the accuracy (%) for individual task, are presented in Tables 3 and 4. The values reported in Table 1 are computed by averaging across tasks within each dataset.

---

[4]`https://platform.openai.com/docs/guides/embeddings#use-cases`

*Table 4.* **Accuracy (%) of embedding models on additional implicit meaning benchmarks. P-Stance** includes stance detection for Trump, Biden, and Bernie. **Implicit Hate Speech (IHS)** comprises detection (Det.), categorization (Cat.), and target identification (Tar.) tasks. The **Social Bias Inference Corpus (SBIC)** consists of target identification (Tar.), intent (Int.), sexism (Sex.), offensiveness (Off.), and bias detection (Bias) tasks. **Political Bias (Pol. Bias)** denotes political ideology classification.

| Model | P-Stance | | | IHS | | | SBIC | | | | | Pol. Bias |
|---|---|---|---|---|---|---|---|---|---|---|---|---|
| | Trump | Biden | Bernie | Det. | Cat. | Tar. | Tar. | Int. | Sex. | Off. | Bias | |
| **Random** | 51.7 | 50.9 | 51.2 | 52.5 | 16.6 | 13.4 | 48.7 | 58.2 | 76.6 | 62.0 | 50.5 | 34.5 |
| **Bag-of-Tokens** (Sun et al., 2025b) | 74.6 | 75.4 | 70.1 | 74.5 | 55.0 | 49.3 | 77.7 | 76.1 | 92.0 | 79.6 | 78.1 | 41.6 |
| *Encoder-only Models* | | | | | | | | | | | | |
| **S-BERT** (Reimers & Gurevych, 2019) | 72.1 | 77.4 | 69.3 | 73.2 | 58.0 | 51.2 | 78.8 | 78.0 | 91.9 | 81.2 | 79.1 | 47.9 |
| **GIST-Small** (Solatorio, 2024) | 76.6 | 78.4 | 72.9 | 74.0 | 59.5 | 51.9 | 77.9 | 77.7 | 93.1 | 81.3 | 78.0 | 49.1 |
| **BGE-Base** (Xiao et al., 2024b) | 74.5 | 78.8 | 69.9 | 74.2 | 60.7 | 53.9 | 78.7 | 78.5 | 92.6 | 82.0 | 78.6 | 52.1 |
| **Angle** (Li & Li, 2024a) | 77.2 | 79.9 | 72.1 | 75.9 | 56.0 | 46.6 | 80.4 | 80.4 | 93.3 | 83.7 | 80.5 | 50.4 |
| **BGE-Large** (Xiao et al., 2024b) | 75.8 | 80.0 | 72.1 | 75.1 | 61.4 | 53.9 | 80.3 | 79.9 | 93.3 | 83.0 | 80.6 | 51.5 |
| **MXBAI-Large** (Lee et al., 2024c) | 76.4 | 78.9 | 71.5 | 76.4 | 56.5 | 47.3 | 80.4 | 80.3 | 93.1 | 83.6 | 80.4 | 50.5 |
| **GIST-Large** (Solatorio, 2024) | 76.8 | 80.1 | 73.2 | 75.0 | 63.1 | 54.4 | 80.5 | 79.3 | 93.3 | 82.9 | 80.9 | 52.5 |
| **Stella** (Zhang et al., 2024a) | 79.2 | 80.0 | 70.2 | 76.9 | 56.7 | 47.6 | 81.2 | 80.9 | 93.4 | 83.2 | 81.5 | 54.4 |
| *LLM-based Embeddings* | | | | | | | | | | | | |
| **Linq-Mistral** (Kim et al., 2024) | 79.4 | 78.8 | 69.3 | 75.1 | 57.8 | 51.2 | 79.7 | 79.1 | 89.8 | 81.9 | 79.8 | 56.8 |
| **E5-Mistral** (Wang et al., 2023; 2022) | 84.8 | 82.3 | 76.1 | 79.2 | 61.7 | 50.9 | 82.0 | **82.5** | 93.3 | 83.9 | 82.1 | **71.5** |
| **GTE-Qwen** (Li et al., 2023b) | 83.8 | 82.0 | 76.9 | 79.1 | 63.2 | 54.5 | 82.1 | 80.6 | 94.0 | 83.7 | 82.0 | 66.8 |
| *Multimodal Encoders* | | | | | | | | | | | | |
| **Jasper** (Zhang et al., 2024a) | 81.6 | 82.6 | 76.2 | 78.4 | 64.6 | 54.1 | 81.4 | 81.2 | 93.9 | 83.1 | 81.5 | 63.9 |
| *Proprietary Embeddings* | | | | | | | | | | | | |
| **OpenAI-Small** | 82.4 | 81.1 | 76.7 | 78.5 | 64.7 | **55.2** | 80.7 | 81.1 | 93.5 | 83.3 | 80.8 | 56.6 |
| **OpenAI-Large** | **87.5** | **83.8** | **79.7** | **80.2** | **67.3** | 53.7 | **82.9** | 82.3 | **94.2** | **84.7** | **83.1** | 66.3 |

**Widespread Variance Across Models** A central observation is that performance varies substantially across both models and tasks, often in ways that are not predicted by surface-level benchmark rankings. For example, many models achieve near-perfect accuracy on *Task 1 (Direct/Indirect Classification)* from PUB, suggesting that some pragmatic cues can be captured by surface correlates. However, several widely used embedding models, such as **GIST-Small**, **S-BERT**, and **BGE-Base**, perform only marginally better than **Bag-of-Tokens** on more challenging tasks, and in some cases fall below it.

This pattern becomes even more pronounced on certain implicature-oriented tasks. On *Task 10 (Implicature NLI)*, multiple models, including OpenAI's proprietary models and the LLM-based models like **E5-Mistral** and **Jasper**, underperform the **Bag-of-Tokens** baseline. This indicates that higher capacity or modern training recipes do not automatically translate into reliable pragmatic competence, and that lexical heuristics can sometimes outperform dense representations when the latter fail to encode the relevant inferential structure. Overall, these results reinforce a key theme of the paper: **Success on conventional surface-level benchmarks does not reliably transfer to tasks requiring deeper interpretive understanding.**

**Strengths of Large and Multimodal Models** While variance is widespread, the results also reveal a consistent advantage for large-scale and multimodal embedding models. **Jasper**, for example, ranks among the top across a wide range of tasks, with particularly strong performance on socially grounded datasets such as IHS and SBIC. Similarly, large-scale models like **E5-Mistral** and **OpenAI-Large** perform strongly across multiple domains, including social bias classification and pragmatic reasoning tasks.

These outcomes suggest that scale and broader pretraining signals can improve robustness for complex semantic phenomena. However, the advantage is not uniform: even high-performing models exhibit sharp drops on specific pragmatic tasks, indicating that capacity helps but does not fully resolve the underlying challenge.

**Persistent Challenges in Implicature and Reference Tasks** Despite improvements from scale and multimodality, certain pragmatic phenomena remain consistently difficult across all evaluated models. In particular, *Task 4 (Implicature Recovery)*

remains difficult across all models, with scores rarely exceeding 50%. Even top-tier models like **GTE-Qwen** and **OpenAI-Large** achieve only modest gains over **Bag-of-Tokens**.

This trend points to a deeper limitation of current training pipelines. Many dominant training signals–whether self-supervised, NLI-based, or retrieval-based–do not systematically require recovering unstated intent or bridging pragmatic gaps. As a result, even powerful models may rely on shallow correlates rather than encoding the inferential structure needed for implicature and reference resolution.

**Implications for Benchmark and Model Design** Taken together, the appendix results expose persistent blind spots in current embedding models, especially for tasks involving implicature, figurative language, presupposition, reference, and socially grounded inference. More importantly, they clarify that these weaknesses are not isolated edge cases: they reflect a systematic mismatch between (i) what embeddings are trained to optimize and (ii) what is required to represent implicit meaning.

Addressing these gaps will likely require both linguistically grounded supervision that targets implicit semantics and benchmarks that directly measure interpretive competence beyond surface similarity. In this sense, the appendix results provide a detailed empirical foundation for the agenda proposed in the paper.

