# OpenReview forum: "Position: Text Embeddings Should Capture Implicit Semantics, Not Just Surface Meaning"
_ICML.cc/2026/Position_Paper_Track — ICML 2026 Position Paper Track regular_

### Official Review · Reviewer_QCD5 · 2026-03-11

**Significance:** 3
**Argument Clarity:** 4
**Rating:** 4
**Confidence:** 4

**Questions:**

Can authors comment on instruction-following information retrieval? It also needs to encode the implicit semantics of either user instructions or corpus, rather than simple surface form matching. It is worth discussing in the paper.

**Alternative Views Section:**

Yes

**Compliance With Llm Reviewing Policy A Conservative:**

Affirmed.

**Discussion Potential:**

3

**Final Justification:**

I did not have strong objections to the publication of this paper.  The authors have answered my questions and I would like to maintain my leaning positive score.

**Paper Summary:**

This paper argues that current text embedding model lacks the ability to encode implicit semantic meaning, which is related to the linguistic explanation such as pragmatics, stance-taking, and sociolinguistics.  The authors also use experiments to demonstrate that current models, regardless if they are encoder-only, LLM-based, multimodal, or proprietary models, all fall short in implicit semantic understanding. The authors argue this issue roots from their lack of diverse training data, dedicated benchmark, and surface-meaning-focused training objective.

**Position:**

Yes

**Position In Title:**

Yes

**Related Work:**

4

**Strengths And Weaknesses:**

This paper is well written, with extensive discussion of past work, including past models, training objectives, model architectures, and benchmarking tools. These studies are sufficient to make a survey paper.

The issue discussed in this paper is important and often omitted by current researchers, because most of the NLP problems can be defined or solved with surface meanings without deep understanding of the implicit meaning.  Capability of implicit semantics is not only a necessary condition for "NLP-complete" models but also a very useful feature for embedding models that can benefit downstream tasks like information retrieval or semantic parsing.

One drawback of this paper is that it blends the benchmarks for embedding models and decoder-only LMs.  There are some tasks that can only be done by embedding models, such as information retrieval and STS. Most of the tasks mentioned in this paper, such as NLI, can be framed to be a generation task, thus can be done with decoder-only LMs, such as a large language model. While embeddings may not convey implicit semantics, LLMs with reasoning traces are much more capable of this.  For these tasks, encoding everything into vectors is a nice-to-have but not necessary properties.

Another weaknesses is that the authors do not propose a clear solution on how to enable embeddings to encode implicit semantics. In section 7, the authors call the researchers to create more diverse datasets, which does not convince me: Researchers have already used nearly all public data to train language models, and adding similar scale diverse is not practical. Also, it is not feasible to collect "implicature, presupposition, stand, and indirect evaluation" data for its high cost to annotate or filter such data.

**Support:**

3

---

> ### Author Rebuttal · Authors · 2026-03-31
>
> We thank reviewer QCD5 for the careful reading and thoughtful feedback. We especially appreciate the reviewer's recognition that the paper is well written, extensively grounded in prior work, and raises an important but often overlooked issue. We are also encouraged by the reviewer's observation that implicit semantics is both relevant to deeper language understanding and potentially useful for downstream embedding applications.
>
> > **(W1) The paper blends benchmarks for embedding models and decoder-only LMs. Some tasks can be solved by LLMs through generation and reasoning traces, so encoding everything into vectors may be nice-to-have rather than necessary.**
>
> Thank you for this important point. We agree that embeddings and decoder-only LMs play different roles and should not be conflated. Our claim is not that embeddings should replace LLM-based reasoning, nor that all tasks requiring implicit understanding must be solved purely in vector space. Rather, our point is that embeddings remain widely used as the first-stage representation in retrieval, clustering, recommendation, reranking, and classifier-based pipelines, where what is preserved in the embedding space directly affects downstream behavior. In this sense, the ability to retain task-relevant implicit semantics is still valuable even if a stronger generative model could solve the same task with full reasoning. We will revise the paper to make this distinction clearer and avoid overstating the role of embeddings relative to decoder-only LMs.
>
> > **(W2) Section 7 does not provide a clear solution. Suggesting "more diverse datasets" is not convincing because raw public data is already saturated and collecting implicit-semantics data is expensive.**
>
> Thank you. We agree that the current discussion should be made more concrete. Our intention is not to suggest simply scaling raw public text further. Instead, the key issue is that embedding models rely on supervision signals such as pairwise similarity, contrastive learning, and retrieval-style objectives, and current supervision rarely makes implicit meaning salient. By "more diverse data," we mean more diverse **training supervision**, not merely more web-scale text. In the revision, we will clarify concrete directions such as: (1) curating smaller but targeted datasets for indirect answers, stance, sarcasm, socially situated language, and related phenomena; (2) constructing contrastive pairs that preserve surface similarity while varying implied meaning; and (3) using LLMs or stronger cross-encoders to synthesize or distill such supervision. We agree that large-scale manual annotation is costly, and we will revise the section to emphasize practical approaches beyond raw data collection.
>
> > **(Q1) Can the authors comment on instruction-following information retrieval? It also requires encoding implicit semantics in user instructions or corpus beyond surface matching.**
>
> Thank you for this excellent suggestion. We agree that instruction-following retrieval is a highly relevant case and should be discussed in the paper. It is indeed a useful example showing that embedding research is already moving beyond simple surface-form matching toward richer forms of semantic conditioning. At the same time, we would view it as an intermediate step rather than a full solution: instruction-following retrieval improves sensitivity to user intent and task framing, but it does not necessarily guarantee robust modeling of broader implicit semantics such as stance, presupposition, or sociocultural meaning. We will add some discussion of this line of work to better position it as a promising bridge between standard surface-semantic embeddings and the richer implicit-semantic representations we advocate.
>
> We appreciate the reviewer's comments and will revise the paper to better distinguish embeddings from generative LMs, make the proposed future directions more concrete, and incorporate instruction-following retrieval into the discussion.

---

> > ### Author Rebuttal · Reviewer_QCD5 · 2026-04-01
> >
> > I did not have strong objection to this paper and the replies have answered all my questions. Thanks for the rebuttal and I will maintain my score.

---

### Official Review · Reviewer_ZpGz · 2026-03-12

**Significance:** 2
**Argument Clarity:** 3
**Rating:** 2
**Confidence:** 4

**Questions:**

1. As the authors note in the Alternative Views section, LLMs are already powerful and capable of handling complex tasks. In this case, is it necessary for embeddings to capture or represent highly complex information?

2. Converting text into embeddings inevitably leads to some information loss. Moreover, embeddings contain limited information, whereas LLMs rely on billions of parameters to encode knowledge. Is it realistic to expect a text embedding to preserve all the information from the original text?

**Alternative Views Section:**

Yes

**Compliance With Llm Reviewing Policy A Conservative:**

Affirmed.

**Discussion Potential:**

2

**Paper Summary:**

The authors claim that the current embedding models mainly focus on surface-level semantics while lack understanding of implicit semantics. The authors suggests the embedding models should be trained on more diverse data and new benchmarks for implicit meaning evaluation should be proposed.

**Position:**

Yes

**Position In Title:**

Yes

**Related Work:**

3

**Strengths And Weaknesses:**

Strengths:

1. The authors provide the foundation of implicit meaning from liguistic theories, which bring backgrounds to readers.

2. The authors well organize the models, training processes and bechmarks of current text embedding research.

3. The authors systematically compared different embedding models on diverse datasets.


Weaknesses:

1. The Figure 1 cannot conclude that the current embedding models are not good at implicit semantics. The authors only compare the average peformance gain over bag-of-tokens. However, if the performance of bag-of-tokens is very high, such as 90, then there is only very small space for the improvement. Also, the selected models are different for surface meaning and implicit semantics, making the comparision unfair.

2. It is well-known that models should be trained on diverse dataset. However, how to obtain such data is challenging. The authors didn't provide potential ways to collect such datasets.

3. The LLMs are usually good at complex reasoning. Do we really need embedding models to capture such implicit semantics?

**Support:**

2

---

> ### Author Rebuttal · Authors · 2026-03-31
>
> We thank reviewer ZpGz for the careful reading and thoughtful feedback. We especially appreciate the reviewer's recognition that the paper provides useful linguistic background, a well-organized overview of current embedding research, and a systematic empirical comparison across diverse datasets.
>
> > **(W1) Figure 1 cannot conclude that current embedding models are not good at implicit semantics. The comparison is based only on average gain over Bag-of-Tokens, which may be affected by ceiling effects, and the selected models differ across surface and implicit evaluations.**
>
> Thank you for this helpful comment. We agree that Figure 1 is not intended as a conclusive or definitive measure of implicit-semantic ability. Rather, it is meant as a teaser that highlights a potential disconnect between strong benchmark performance on surface meaning and more limited improvement on tasks requiring implicit understanding. We agree that gains over Bag-of-Tokens depend on the lexical baseline and available headroom, and that the use of different model sets makes the comparison less strict. Regarding the use of different model sets, we explicitly selected the top 3 performing models from each respective side (surface meaning vs. implicit semantics). Our goal was to compare the "best-case" performance ceilings in both domains. By showing the strongest possible contenders for each category, we aimed to demonstrate that even the absolute best models struggle to achieve proportional gains on implicit tasks. We will revise the paper to clarify these limitations and present Figure 1 more carefully as an illustrative motivation, while emphasizing that our main argument rests on the broader empirical and conceptual analysis in the paper.
>
> > **(W2) It is well known that models should be trained on diverse datasets, but the paper does not explain how such data can be obtained.**
>
> Thank you. We agree that this part should be made more concrete. Our point is that embedding models require a different kind of supervision than autoregressive LLMs, which limit the acquisition of annotated data. Generative models can learn from raw text alone, whereas embedding models are typically trained with pairwise, contrastive, or retrieval-style objectives. For implicit semantics, this requires data where pragmatic, attitudinal, or sociocultural distinctions are reflected in the supervision signal. In the revision, we will expand this discussion with concrete directions, including: (1) targeted curation from dialogue, indirect answers, sarcasm, stance-rich discourse, and socially situated language; (2) contrastive pair construction that keeps surface meaning similar while varying implied meaning; and (3) LLM-based synthetic or relabeled data generation guided by linguistic theory.
>
> > **(W3 / Q1) If LLMs are already strong at complex reasoning, do we really need embeddings to capture implicit semantics?**
>
> Thank you for raising this important question. We agree that LLMs are powerful and that embeddings are not meant to replace them. Our point is that embeddings are still widely used as **feature representations** in downstream systems, including classification, clustering, retrieval, recommendation, and filtering. For example, practitioners often use embeddings as input features to train classifiers or as representations in recommendation/ranking systems, which is also how current embedding research commonly evaluates them. In such settings, what is preserved in the embedding space directly shapes downstream behavior. If important implicit signals such as stance, intent, or social framing are not retained, downstream models built on these features may also miss them. Thus, our goal is not for embeddings to match the full reasoning capacity of LLMs, but for them to better preserve task-relevant implicit information.
>
> > **(Q2) Is it realistic to expect an embedding to preserve all information from the original text?**
>
> We fully agree that it is neither realistic nor necessary for a single embedding to preserve *all* information from the source text. Our goal is not lossless semantic compression. Rather, the key question is which information is preserved and optimized for. Current embedding objectives strongly favor surface-level semantic similarity and relevance. Our argument is that some aspects of implicit meaning, such as stance, indirect intent, and socially situated framing, are also important for many applications, yet remain weakly represented in current embedding spaces. We will revise the paper to make clear that our proposal is not to encode all possible information, but to broaden the modeling target beyond surface semantics toward representations that retain more task-relevant implicit information.
>
> We appreciate the reviewer's comments and will revise the paper to better qualify the role of Figure 1, make the data discussion more concrete, and clarify the practical motivation for embeddings that better preserve implicit semantics.

---

> > ### Author Rebuttal · Reviewer_ZpGz · 2026-04-01
> >
> > Thanks for the authors' response. However, I still have the following concerns/suggestsion:
> >
> > 1. I suggest the authors update the Table 1 to make a fair comparision.
> >
> > 2. The authors claim "practitioners often use embeddings as input features to train classifiers or as representations in recommendation/ranking systems". However, there is no evidence that the recommendation/ranking systems require such implicit meanings.
> >
> > 3. How does the authors define  task-relevant implicit information for different tasks?

---

### Official Review · Reviewer_pLEK · 2026-03-13

**Significance:** 3
**Argument Clarity:** 3
**Rating:** 4
**Confidence:** 2

**Questions:**

Please discuss the cases where embedding models perform well on the PUB tasks and clarify the mechanisms that might explain their success.

**Alternative Views Section:**

Yes

**Compliance With Llm Reviewing Policy A Conservative:**

Affirmed.

**Discussion Potential:**

4

**Paper Summary:**

Through a comprehensive analysis of current text embedding models and their limitations, this paper argues that these models — including LLM‑based ones trained primarily on surface‑level semantic objectives — fail to capture implicit semantics. The claim is evaluated via a pilot study on seven benchmark datasets designed for implicit‑semantics evaluation. Experimental results demonstrate these models’ limited ability to capture implicit meaning, which supports the paper’s central claim.

**Position:**

Yes

**Position In Title:**

Yes

**Related Work:**

4

**Strengths And Weaknesses:**

**Strengths**
1) The paper is well structured and presents an engaging position.
2) The pilot study—comparing a lexical Bag‑of‑Tokens baseline with a range of state‑of‑the‑art text embedding models—effectively highlights the importance of the authors’ argument.

**Weakness**
The paper reports that a lexical Bag‑of‑Tokens baseline sometimes outperforms surface‑semantic embedding models on certain three‑tier implicit‑semantics tasks, illustrating embedding models’ limitations. However, this pattern is not consistent across all PUB tasks: Tables 3–4 show that embedding models achieve noticeable gains on the majority of the 14 PUB tasks. A more detailed discussion of these positive results would strengthen the paper’s claim.

**Support:**

3

---

> ### Author Rebuttal · Authors · 2026-03-31
>
> We thank reviewer pLEK for the careful reading and encouraging feedback. We especially appreciate the reviewer's recognition that the paper is well structured and that the pilot study helps highlight the importance of the paper's central argument.
>
> > **Weakness / Question: Please discuss the cases where embedding models perform well on the PUB tasks and clarify the mechanisms that might explain their success.**
>
> Thank you for this important point. Our intention is not to claim that current embedding models completely fail on all implicit-semantics tasks. Rather, our claim is that their gains are uneven and remain limited relative to their strong performance on standard surface-semantic benchmarks.
>
> In the PUB results, embedding models do show noticeable improvements on several tasks, especially when the implicit meaning is closely tied to **local lexical, syntactic, or discourse cues**. In such cases, models may benefit from large-scale pretraining, instruction tuning, and stronger contextual encoding, which allow them to recover some pragmatic regularities even without being explicitly trained for implicit semantics. This is particularly plausible for tasks where the inference is relatively conventionalized or where surface patterns correlate strongly with the underlying pragmatic label.
>
> At the same time, these positive results do not contradict our main argument. Instead, they suggest that current embeddings capture **some shallow or partially lexicalized forms of implicit meaning**, while still struggling with more difficult cases that require richer contextual grounding, speaker modeling, or socially situated interpretation. This is why the overall performance remains far from robust and why the gains over lexical baselines shrink substantially compared with standard embedding benchmarks.
>
> In the revision, we will expand the discussion of PUB to make this nuance clearer: current embedding models are not uniformly poor on all implicit tasks, but their success is selective and tends to occur when implicit meaning is more recoverable from surface-accessible cues. We will clarify that this pattern is consistent with our broader claim that existing embedding objectives do not yet support a stable and general representation of implicit semantics.

---

> > ### Author Rebuttal · Reviewer_pLEK · 2026-04-06
> >
> > Thank you to the authors for their detailed response. As I had already given the paper a positive score, I keep my recommendation and score unchanged.

---

### Official Review · Reviewer_vXTn · 2026-03-20

**Significance:** 3
**Argument Clarity:** 3
**Rating:** 5
**Confidence:** 4

**Questions:**

1), It is widely acknowledged that most Large Language Models (LLMs) are already pre-trained on web-scale corpora that encompass nearly all available human-generated data. Given this saturation, the authors should clarify how researchers can practically collect 'more diverse' training data in the future. What specific sources or modalities are currently missing from these massive datasets that would effectively capture implicit semantics?

2), While the paper argues that current training objectives (e.g., Self-Supervised Learning and Supervised Learning) fail to capture implicit embeddings, the discussion in Section 7.3 remains too abstract for readers to grasp the proposed alternative. The authors need to explicitly define what the 'right' modeling goal looks like

**Alternative Views Section:**

Yes

**Compliance With Llm Reviewing Policy A Conservative:**

Affirmed.

**Discussion Potential:**

3

**Final Justification:**

After carefully reviewing the other comments, I recommend that the authors incorporate the discussion from the rebuttal phase into the final revised draft.

**Paper Summary:**

This position paper argues that current embedding models focus predominantly on surface-level meaning (such as semantic similarity inferred from lexical relevance) rather than implicit semantics. The authors propose the concept of implicit embeddings, defined as contextual representations that consider pragmatic inference, speaker intent, and sociocultural context. Specifically, the paper first explicates implicit meaning at the utterance, speaker, and societal levels, then discusses the limitations of existing embedding models. Finally, the authors analyze the underlying causes of these limitations regarding datasets, training loss, and benchmarks, offering insights on how to capture implicit embeddings in these areas.

**Position:**

Yes

**Position In Title:**

Yes

**Related Work:**

3

**Strengths And Weaknesses:**

Strengths:

1), The paper provides a detailed review of current embedding models, covering model architectures, training processes, and evaluation benchmarks. This offers readers a clear and solid understanding of the current landscape.

2), The authors provide compelling reasons explaining why current models fail to capture implicit meaning, specifically analyzing issues related to datasets, training objectives, and benchmarks. The discussion is logical, clear, and well-supported.

Weaknesses:

1), While the three-level framework for implicit meaning (Utterance, Speaker, and Society) is intuitive, the definitions provided are somewhat vague and lack operational rigor. Without clearer definitions, it may be difficult for future studies to build upon this work or develop concrete solutions.

2), Given that LLM-based embedding models trained on web-scale corpora demonstrate strong understanding and reasoning capabilities, the authors should elaborate on why these models still fail to capture implicit semantics. The current explanation relies too heavily on inference cost (Lines 212–214); a deeper analysis of the architectural or objective function limitations is needed.

3), The suggestions such as "Curating More Diverse Training Data" and "Designing Benchmarks for Implicit Meaning" are intuitive to the LLM community, and the "Framing implicit Semantics as a Modeling Goal" too abstract. The authors should provide a more concrete definition of this modeling goal or suggest specific methodological approaches to make this direction actionable.

**Support:**

3

---

> ### Author Rebuttal · Authors · 2026-03-31
>
> We thank reviewer vXTn for the careful reading and encouraging feedback. We especially appreciate the reviewer's recognition that the paper provides a clear overview of the embedding landscape and a well-supported analysis of why current models struggle with implicit meaning. We are glad that the reviewer found these aspects valuable, and we will further strengthen the paper by addressing the concerns below.
>
> > **(W1) Definition of the 3-level framework of implicit meaning**
>
> Thank you for raising this point. Our goal is not to fully define *implicit meaning* as a linguistic concept, which is itself a long-standing challenge in linguistics. Rather, the proposed 3-level framework is intended as a linguistically grounded summary that helps the NLP community reason about where implicit meaning may arise in text. In particular, the framework highlights three complementary aspects of interpretation: the **utterance level** (e.g., pragmatics, implicature, presupposition), the **speaker level** (e.g., stance, evaluation, commitment), and the **society level** (e.g., sociocultural meaning, identity, ideology, and social indexicality). In the revision, we will clarify that this framework is an analytical lens rather than a strict ontology, and we will make the definitions more explicit and operational.
>
> > **(W2) Why LLM-based embedding models still lack implicit meaning**
>
> Thank you for this important point. We agree that inference cost alone is not a sufficient explanation. Our main argument is that many LLM-based embedding models are obtained by converting generative models into embedders through continued fine-tuning on embedding objectives. This alignment step mainly rewards surface semantic similarity, retrieval relevance, or paraphrastic equivalence. As a result, even if the underlying generative model may contain stronger reasoning capabilities, the embedding adaptation process does not explicitly preserve implicit semantics in the final representation space. This is also consistent with our experiments: LLM-based embedding models are stronger than encoder-only baselines, but there is still substantial room for improvement on implicit semantics tasks.
>
> > **(W3, Q2) Definition of "Framing Implicit Semantics as a Modeling Goal"**
>
> Thank you for pointing this out. We agree that this goal sounds abstract in the current draft. What we intend is to call for two concrete shifts:
> (1) treating implicit semantics as an explicit training target rather than an incidental byproduct of surface-semantic learning; and
> (2) developing model structures and objectives that preserve information relevant to implicit meaning.
>
> Concretely, this could include training losses that distinguish texts with similar surface meaning but different implied meanings, multi-task supervision over pragmatics/stance/social meaning, or architectures that better retain socially and pragmatically grounded information. We will revise this section to make the modeling goal more explicit and actionable.
>
> > **(Q1) Why do we still need more diverse data for embedding model training?**
>
> Thank you for this question. The key issue is not simply the total amount of human-generated text available on the web, but the kind of supervision needed to train embedding models. Autoregressive generative models can learn from raw text alone, whereas embedding models typically rely on pairwise or contrastive supervision. For implicit semantics, this requires high-quality examples where the relevant pragmatic, attitudinal, or sociocultural distinction is preserved in the supervision signal. Such data is much harder to obtain through naive web crawling. Therefore, by "more diverse data," we mean more linguistically and socially diverse supervision, including curated or synthetic pairs that reflect implicature, presupposition, stance, indirectness, social framing, dialectal variation, and related phenomena.
>
> > **(Q2) Define what the "right" modeling goal looks like**
>
> Thank you for asking for a clearer formulation. In our view, a modeling goal is to learn embeddings that are not only useful for surface-level semantic similarity, but that also preserve information needed for downstream tasks requiring implicit understanding. In other words, a strong embedding should serve as a high-quality feature representation for tasks such as stance-aware retrieval, socially grounded classification, recommendation, clustering, or other applications where implied meaning matters. We will revise the paper to define this goal more explicitly and connect it to concrete downstream capabilities.
>
> We appreciate the reviewer's feedback and will make sure these clarifications are integrated into the revision.

---

> > ### Author Rebuttal · Reviewer_vXTn · 2026-04-04
> >
> > Thank you for your response. I will change the rating to “Accepted.”
> >
> > In the revised draft, the author should provide a detailed discussion of the scale of diverse training data and the guiding principles for training objectives.

---

### Decision · Program_Chairs · 2026-04-30

**Decision:**

Accept (regular)

**Comment:**

The paper raises an important and under-discussed issue in embedding research, and several reviewers found the central position timely and valuable. Its main contribution is to make explicit a limitation that is often glossed over in current embedding work: strong performance on surface-semantic benchmarks does not imply robust handling of pragmatic, attitudinal, or socially grounded meaning.

The reviews were overall positive, and most concerns were addressed well in rebuttal. In particular, the authors clarified the intended role of the three-level framework, explained more carefully why LLM-based embedders may still miss implicit semantics, and made the proposed future directions more concrete. I do note that one reviewer remained unconvinced, mainly on the scope of the empirical evidence and the practical necessity of embeddings capturing richer implicit signals. I think these are fair limitations to note, but on balance they do not outweigh the paper’s value as a position paper.